# Bridging the Knowledge Gap: A National Survey on MASLD Awareness and Management Barriers in the Saudi Population

**DOI:** 10.3390/healthcare13243322

**Published:** 2025-12-18

**Authors:** Abdulrahman Alwhaibi, Wael Mansy, Wajid Syed, Salmeen D. Babelghaith, Mohamed N-Alarifi

**Affiliations:** Department of Clinical Pharmacy, College of Pharmacy, King Saud University, Riyadh 11451, Saudi Arabia; aalwhaibi@ksu.edu.sa (A.A.); wali@ksu.edu.sa (W.S.); sbabelghaith@ksu.edu.sa (S.D.B.); malarifi@ksu.edu.sa (M.N.-A.)

**Keywords:** MASLD, KAP, barriers, Saudi Arabia

## Abstract

Background: Metabolic dysfunction-associated steatotic liver disease (MASLD) is the leading cause of chronic liver disease worldwide. It greatly increases hepatic cirrhosis and cancer, cardiovascular disease, and chronic kidney disease. Despite the rising frequency of MASLD in Saudi Arabia, public understanding of its management is lacking. Objective: This study seeks to evaluate public knowledge, attitudes, and management barriers related to MASLD, thereby informing future educational and preventive strategies. Methods: A cross-sectional study was conducted from November 2023 to October 2024, involving 502 participants across Saudi Arabia, employing a modified self-administered online questionnaire. Data was analyzed using SPSS 25. Descriptive statistics and Chi-square tests were used to investigate correlations between knowledge or attitude levels and demographics, with a significance threshold of *p* < 0.05. Results: Less than half of the respondents who took part (47.2%) had heard of MASLD. Of them, 24.9% had good knowledge, 38.2% had fair knowledge, and 36.9% had low understanding. There were strong links between knowledge and age, education, and job status, but not between knowledge and gender (*p* = 0.514). People were somewhat aware that being overweight (48.4%) and having high cholesterol (51.8%) were risk factors, but they often had wrong ideas regarding diabetes and high blood pressure. Only 7.8% of those surveyed said they had been formally diagnosed, and 74.4% of those who had been were given advice on how to change their lifestyle. Barriers to management included the idea that lifestyle change alone suffices (46.7%), the absence of medical advice (46.7%), and insufficient disease awareness (33.3%). Conclusions: The research shows that many Saudis are unaware of MASLD and have misconceptions about it. Targeted health education programs, greater provider–patient communication, and primary care MASLD knowledge are needed to close these gaps and promote disease prevention and management.

## 1. Introduction

Metabolically dysfunction-associated steatotic liver disease (MASLD), previously referred to as nonalcoholic fatty liver disease (NAFLD), is a widespread chronic liver condition, impacting about 30% of the global population. MASLD is a multifactorial condition arising from intricate interactions among many cardiometabolic and environmental risk factors [1,2]. However, the geographical distribution of the disease prevalence differs among countries [3]. Africa has a lower prevalence of MASLD than South America and the Middle East, followed by Asia, the United States, and Europe. MASLD was seen in 16.8% (11.1–22.5%) of the adult population in Saudi Arabia. In the Kingdom of Saudi Arabia, the total prevalence of MASLD among adults was 16.8% (11.1–22.5%) [4]. MASLD is considered a public health issue, as it has been identified as the main cause of chronic liver disease globally [5]. In addition to having the potential to cause cirrhosis and liver cancer [6], MASLD is a known risk factor for cardiovascular diseases [7,8], type 2 diabetes [9], and chronic kidney disease [10]. According to several studies, those with MASLD had a higher chance of dying from all causes than those without [11,12]. In particular, compared to those without NAFLD, patients with NAFLD have a 30–60% increased risk of dying from cardiovascular disease (CVD) and cancer [3,6,8].

Several studies explored the general population’s awareness and understanding of MASLD, and found it inadequate, with a significant knowledge gap. According to Singh and his colleagues, people suspected of having MASLD had relatively low knowledge of the disease. Furthermore, 18% of individuals with high metabolic risk—such as those who were overweight or obese and/or had insulin resistance—were aware of non-alcoholic fatty liver disease [13]. Another study conducted in Malaysia revealed that 21.4% of all the respondents knew the condition’s risk factors, screening test, and effects of MASLD [14]. According to national surveys conducted by Someili and his colleagues among general adults in Jazan Province, aimed at evaluating knowledge, attitudes, and influencing factors related to MASLD, they found that 226 (41.9%) of participants had fair knowledge of MASLD, whereas the majority (244, 45.2%) had poor knowledge [15]. A study in Riyadh, Saudi Arabia, surveyed 397 people to assess their perception, attitude, and beliefs about MASLD. Most respondents had inadequate knowledge of MASLD [4]. Additionally, numerous studies from around the world have demonstrated that physicians are aware of MASLD, undervalue the disease’s prevalence and related dangers, and lack knowledge of the disease’s proper examination [16,17]. Furthermore, numerous studies have investigated the current management status of the disease and the challenges faced by the general population. Studies indicate that patients diagnosed with MASLD, even with an understanding of their condition, fail to make substantial behavioral modifications [18].

Improving understanding and attitudes around NAFLD requires targeted educational programs for professional and academic groups, particularly women and individuals with low incomes, along with sustained efforts to ensure this increased awareness translates into meaningful action to address the knowledge gap and foster positive behavioral changes. On 14 March 2024, the U.S. Food and Drug Administration announced the approval, under the accelerated approval pathway, of Rezdiffra^®^ (resmetirom) to help adults with noncirrhotic nonalcoholic steatohepatitis (NASH) complicated with moderate to severe liver scarring (fibrosis). NASH happens when nonalcoholic fatty liver disease (NAFLD) gets worse, causing long-term liver inflammation that can eventually lead to liver scarring [19]. The American Gastroenterological Association (AGA), American Association for the Study of Liver Diseases (AASLD), and European Association for the Study of the Liver (EASL) endorse lifestyle modifications that achieve a 5–10% reduction in body weight, emphasizing strength training, aerobic exercise, and hypocaloric diets [20,21,22].

If physicians do not comprehend the public’s challenges and needs regarding the management of MASLD, achieving therapeutic objectives will be difficult. To address the disconnect between awareness, knowledge, and the management of MASLD, it is essential to investigate the needs and barriers faced by the general population concerning MASLD, along with their awareness and understanding of the disease [16].

Currently, limited research exists in Saudi Arabia assessing the knowledge and attitudes of the adult population regarding MASLD, with findings indicating low levels of awareness. Previous studies in Saudi Arabia did not examine the barriers to management, which can involve insufficient physician guidance, low health literacy, lifestyle challenges, and financial constraints. Understanding the level of knowledge and attitudes, as well as the challenges patients face in managing MASLD, is crucial due to the potential for severe complications, including cirrhosis and liver cancer. This study seeks to evaluate public knowledge, attitudes, and management barriers related to MASLD, thereby informing future educational and preventive strategies.

## 2. Methods

### 2.1. Study Setting

This cross-sectional study was carried out in Saudi Arabia between April to October 2024. It aimed to assess the knowledge, attitudes, and management barriers of MASLD among the public in Saudi Arabia. The ethics committee at King Saud University accepted this study for human research. Additionally, before data collection, informed consent was gathered from the participants, who were assured that their data would be exploited solely for research purposes and that anonymity would continue to apply throughout the project. The sample size was determined with Raosoft software (Raosoft Inc., Seattle, WA, USA), targeting a 95% confidence level and a 40% expected response distribution with a ±5% margin of error. The requisite minimum sample size was established at 369 participants. The planned sample size was augmented to 500 participants [23].

### 2.2. Study Tool

The study adopted a validated, systematic, self-administered online questionnaire modified from previous studies [1,15]. Participants were informed to complete the questionnaire independently at their convenience. We recruited the adult Saudi population—specifically, individuals aged 18 and older who are fluent in both Arabic and English—and excluded those who refused to participate, non-Saudis, and individuals under 18 years of age. While developing the questionnaire, the authors agreed to use the term NAFLD instead of MASLD, since the former is more familiar at the national level in KSA, despite MASLD being the current term used in the literature. The questionnaire comprised four parts. The first part included demographic data such as gender, age, and education levels, job, and body mass index. The second part assessed the knowledge of participants about NAFLD utilizing 11 questions (yes/no) covering risk factors, complications, and prevention. The third part consisted of six questions (yes/no) that assessed participants’ attitudes toward the disease regarding the causes, seriousness, and potential health outcomes. The fourth part included six items assessing the current NAFLD management practice and barriers to assessment for individuals diagnosed with NAFLD, such as the need to change their lifestyle, physician counseling, or issues with cost, time, or disease awareness. The aggregate responses to each question were utilized to calculate the overall knowledge score. Participants are categorized based on their knowledge levels: those who correctly answered up to 30% of the knowledge items are classified as having poor knowledge, those who answered between 31% and 60% are deemed to have fair knowledge, and those who answered more than 60% are considered to have good knowledge. A comparable methodology was employed to assess participants’ attitudes toward NAFLD. The responses were assessed and categorized as negative (poor attitude), neutral (fair attitude), or positive (good attitude) based on the percentage of affirmative answers. The internal consistency of the knowledge and attitude tiers was assessed using Cronbach’s alpha, yielding values of 0.72 and 0.81, respectively.

### 2.3. Data Analysis

SPSS version 25 was used for statistical analysis. Descriptive statistics, including frequencies and percentages, were used to summarize demographic data, knowledge, attitude, and barriers. Chi-square tests were used to examine relationships between knowledge or attitude scores and demographic factors (age, gender, education, and job). For all analyses, *p* < 0.05 was considered statistically significant. Appendix A provided shows the data collected and analysis.

## 3. Results

Five hundred two individuals completed the questionnaire. Females constituted slightly over half of the participants (54%), whereas males comprised 46% of the sample. The age distribution of participants indicated that the predominant group (51.2%) fell within the 18–24 years range, followed by individuals aged 34–51 years (24.3%), 52–64 years (11.4%), and 25–33 years (10.6%). Regarding employment status, a significant portion (47.8%) of the participants were students. Nine percent were either unemployed or retired, while 34.3% were employed. An examination of participants’ self-reported body mass index (BMI) revealed that 45.8% were classified within the normal range. Only 6.8% of the participants were obese, and 29.5% were overweight. 10.8% of respondents were unaware of their BMI. These results are shown in Table 1.

### 3.1. Knowledge of Participants About NAFLD

About 47.2% of participants reported that they had heard of the term NAFLD. When asked whether NAFLD can occur without alcohol intake, 45% of participants correctly answered “Yes.” About 14% of participants thought NAFLD cannot occur without alcohol, and a large portion, 41.2%, indicated, “I don’t know.” Approximately two-thirds of participants believed NAFLD is a condition that requires hospital treatment. The findings indicate that the majority of participants (78.9%) acknowledge NAFLD as a potentially problematic condition, which is positive and shows that they are cognizant of how dangerous it is. In terms of specific complications, 64.1% of participants identified liver cirrhosis as a possible outcome, followed by liver cancer (42.8%) and vascular diseases (29.7%). In addition, the majority of participants (82.7%) knew the common risk factors for NAFLD. The most frequently reported were high cholesterol (51.8%) and obesity (48.4%). Other risk factors identified included lack of physical activity (40.2%), intake of fatty foods (38.4%), liver tumor (35.7%), diabetes (29.1%), and hypertension (23.9%). Notably, 27.3% of participants stated that they did not know the causes of fatty liver disease. According to the results, a significant percentage (42.2%) of participants are still unsure, even though slightly more than half acknowledge NAFLD as a potentially life-threatening condition. When asked how NAFLD is diagnosed, 47.6% of participants correctly answered “All of the above,” demonstrating that there are several diagnostic techniques. As diagnostic methods, blood tests (9.8%), body mass index (BMI) evaluation (8%), and ultrasound imaging (8%) were less commonly mentioned individually. Interestingly, 26.5% of participants said they did not know of the diagnostic techniques. These results are displayed in Table 2.

Moreover, this study assessed the participants’ knowledge about the management strategy for non-alcoholic fatty liver disease, as shown in Table 3. The participants revealed several approaches to managing NAFLD. Dietary changes, cutting back on processed foods, sweets, and saturated fats, and increasing physical activity, were the most often reported strategies (66.5% and 66.5%, respectively). The significance of consuming a balanced diet rich in fruits, vegetables, fiber, healthy proteins, and whole grains was also acknowledged by 60.8% of participants. A small percentage (30.9%) acknowledged the role of diet or medication in reducing cholesterol and triglyceride levels, while an equivalent percentage (30.9%) of participants indicated that vitamins might serve as a treatment strategy.

The distribution of the study participants’ knowledge about NAFLD is as follows: 25% have good knowledge, 38% have fair knowledge, and 37% have poor knowledge. These results indicate that almost two-thirds of the participants (63%) had fair to good knowledge. Figure 1 shows the participants’ knowledge level regarding NAFLD.

Analysis of the association between sociodemographic variables and knowledge levels about NAFLD revealed no significant relationship between gender and knowledge levels (*p* = 0.508). This indicates that both males and females in our study demonstrated comparable levels of awareness. However, significant associations were noted with educational level, age, and job status. Those who were working, had postgraduate degrees, and were between the ages of 20 and 40 had higher levels of NAFLD knowledge than their peers (*p* < 0.05). These results are displayed in Table 4.

### 3.2. Attitude of Participants About NAFLD

Participants’ attitudes on the causes and complications of NAFLD are shown in Table 5. About half of the participants (48.4%) believed that obesity causes NAFLD. Only 29.1% of participants thought diabetes was a contributing factor to NAFLD, while 70.9% disagreed. Similarly, 76.1% disagreed with the statement that hypertension affects NAFLD, whereas only 23.9% agreed. Regarding complications, 42.8% of participants identified liver cancer as a potential outcome of non-NAFLD, whilst 57.2% did not. Of those surveyed, 51.8% thought that elevated blood cholesterol was a contributing cause to NAFLD, while 48.2% disagreed. Only 29.7% of people were aware that NAFLD was associated with cardiovascular disease, and 70.3% did not. Furthermore, 40.2% of participants thought that lack of physical activity contributes to NAFLD.

However, the analysis of the association between sociodemographic variables and attitude levels about NAFLD showed no significant relationship (*p* > 0.05), as shown in Table 6.

### 3.3. Current NAFLD Management Status and Barriers

Table 7 presents responses regarding the status of management and obstacles to NAFLD management. The majority of individuals (92.2%) had not received a diagnosis of NAFLD, whereas only 7.8% reported a formal diagnosis. Among individuals diagnosed and hospitalized, 74.4% reported receiving lifestyle adjustment counseling, whereas 25.6% did not receive such guidance. Regarding follow-up, 15 participants (38.5%) did not pursue additional treatment, whereas 24 participants (61.5%) underwent further hospital testing and management. The primary reasons for not following up (n = 15) included the belief that lifestyle changes alone could adequately manage the condition (46.7%) and the absence of physician recommendations indicating the necessity for additional management (46.7%). Barriers included insufficient awareness of the severity of fatty liver disease (33.3%), medical expenses (26.7%), and time constraints (13.3%).

Regarding prevention and management behaviors during hospital follow-up (n = 24, 61.5%), 20.8% cut calories, 29.2% increased physical activity, and 41.7% reported using weight reduction approaches. Pharmacological and supplemental treatment includes hospital-prescribed medications for liver disease or hyperlipidemia (25.6%) and supplements ordered online or from pharmacies (20.8%). Remarkably, 33.3% said they did not take any action to manage their NAFLD.

### 3.4. Perceptions on Long-Term Management and Program Participation

This study also evaluated the perception of the most important aspects of long-term management (multiple responses allowed), as shown in Table 8. Most participants viewed making time for lifestyle modifications as the most crucial component of successful long-term therapy of NAFLD (63.7%). This was followed by providing nutritional advice and periodic management by a nutritionist (57%), instructions on proper diet and exercise from a physician (56.4%), advice on exercise and periodic management by a sports specialist (46.2%), and consideration of health with treatment costs (44%). Regarding engagement with a mobile app, 34.3% strongly agreed and 9.4% agreed that they would use a mobile app to prevent or manage NAFLD, while 24.3% were neutral, 12.2% disagreed, and 19.9% strongly disagreed. Similarly, willingness to participate in a public health program showed that 34.3% were willing to actively participate, 9.4% were willing to participate, 24.3% were neutral, 12.2% expressed little interest, and 19.9% reported no interest in attending such programs.

## 4. Discussion

Since MASLD is becoming a serious public health concern, it is closely linked to metabolic disorders and has the potential to progress to cirrhosis or hepatocellular cancer. Although these risks exist, the disease is often underdiagnosed and undertreated, especially in Asian regions such as Saudi Arabia, where lifestyle-related risk factors are common [24]. Therefore, it is crucial to investigate knowledge, attitudes, and management barriers to understand how people perceive the disease, adhere to medical information, and use preventative or treatment strategies.

In this study, awareness of MASLD was moderately low; just 47.2% of participants said that they had heard of NAFLD. This level of knowledge is notably lower than that reported in some studies from Saudi Arabia. For example, a study in the Riyadh region reported that 79.4% of participants knew of MASLD, indicating significantly greater public awareness [4]. Our findings were marginally greater than those reported in the Jazan and Taif regions of Saudi Arabia (38% and 36.6%, respectively) [25,26]. In an international comparison, our findings differ from those of a Korean study, which reported that 72.8% of participants were aware of MASLD [1]. Discrepancies in results arise from variations in healthcare access, health education initiatives, demographic factors, and lifestyle choices. The limited public awareness of MASLD, in specific regions, highlights the need for focused educational programs to improve understanding and promote earlier detection and management of the condition [27,28].

The current study indicates that knowledge distribution regarding NAFLD shows that 24.9% of participants possess outstanding knowledge, 38.2% have fair knowledge, and 36.9% exhibit poor knowledge. These results indicate a notable disparity requiring attention, with approximately two-thirds of respondents demonstrating fair to good awareness, while over one-third exhibited insufficient understanding. A recent national study revealed that 23.5% of participants had poor knowledge, 61.4% had fair knowledge, and only 15.2% had satisfactory knowledge [23], but our study revealed that a greater proportion of individuals demonstrated excellent knowledge (24.9% compared to 15.2%), suggesting a higher level of awareness. The current results found a higher percentage of individuals with poor knowledge (36.9%) compared to the other national one (23.5%). Contrary to these two studies, a survey in the province of Jazan found that 45.2% of participants exhibited poor knowledge, while 41.9% demonstrated fair knowledge, suggesting a lower level of awareness [14]. These disparities in NAFLD awareness across Saudi Arabia, potentially attributable to differences in educational attainment, access to medical care, or health promotion initiatives. A population from Hong Kong also showed a lack of knowledge regarding MASLD [29]. An American study revealed that the majority of individuals, irrespective of demographic factors, lacked knowledge regarding MASLD [30].

Additionally, in this study, there was no significant relationship between gender and knowledge levels (*p* = 0.508). This indicates that both males and females in our study demonstrated comparable levels of awareness. Nonetheless, previous studies in Saudi Arabia have reported statistically significant gender differences in MASLD knowledge [4,15,22]. Conversely, our results showed a strong correlation between MASLD knowledge levels and age, employment status, and educational levels. Participants aged 20 to 40, employed or holding postgraduate degrees, showed noticeably greater levels of knowledge than their peers. These findings are consistent with several Saudi Arabian and worldwide studies that have emphasized the importance of education, younger age groups, and work status in influencing MASLD awareness [4,15,23,26].

The current study reveals significant misconceptions and gaps in participants’ attitudes toward the causes and complications of NAFLD. Only 48.4% correctly identified obesity as a risk factor for MASLD while 51.8% thought elevated blood cholesterol was a contributing factor. This demonstrates some positive attitude toward realizing how obesity and elevated blood cholesterol contribute to MASLD, but it remains suboptimal given that obesity and dyslipidemia are among the most well-known and modifiable risk factors for MASLD [18,30,31]. Comparing these levels of attitudes to previous national studies, it is comparatively low. According to a study conducted in the province of Jazan, for example, the majority of participants (84.4%) acknowledged obesity as a contributing factor to MASLD. Despite the high incidence of overweight and obesity being prevalent in 32.8% and 23% of the population, respectively, in the Kingdom [32], the comparatively lower recognition in our study points to ongoing inadequacies in health education and awareness programs.

In our study, the majority of participants (70.9%) did not identify diabetes as a contributing factor to MASLD. Similarly, more than 75% of participants did not identify that hypertension is a risk factor for MASLD. These results reveal a critical gap in public awareness, given that both type 2 diabetes mellitus and hypertension are well-established metabolic risk factors for the development and progression of MASLD. It is well known that a high risk factor for the development of MASLD is the metabolic syndrome, which involves diabetes and hypertension. Insulin resistance, compensatory hyperinsulinemia, central abdominal obesity, and a decrease in high-density lipoprotein cholesterol are the causes that might contribute to this chronic liver disease [33].

Many studies reported that primary management of MASLD, lifestyle modification, includes weight loss achieved through diet with or without exercise [18,30]. In support of these findings, our results indicated that just 7.8% reported having a formal diagnosis of MASLD. Of those who received a diagnosis and visited the hospital, the majority (74.4%) reported receiving counseling to adjust their lifestyle. This high percentage indicates that healthcare professionals are appropriately emphasizing dietary and lifestyle changes. A Korean study found that 59.3% of patients reported counseling or advice on lifestyle modification [1].

The current data showed that the belief that lifestyle changes alone were adequate for disease control (46.7%) and the absence of physicians’ advice indicating the need for additional management (46.7%) were the most commonly cited reasons for not following up with hospital visits among patients with MASLD. Underestimating the severity of MASLD (33.3%) was another obstacle. The results indicate that adherence to follow-up treatment is significantly affected by patient perceptions and deficiencies within the healthcare system. Our findings are in line with a Korean study where 50.6% of participants thought they could manage the disease on their own by changing their lifestyle (e.g., controlling their weight or exercise), and 33% of participants stated that their doctor had never told them that disease management was necessary [1]. The similarity between our results and those from Korea points to a problem that affects all populations; many patients believe that changing their lifestyle is sufficient on its own, and inadequate doctor counseling feeds this belief. These results together call attention to two crucial interventions: improving patient education to correct misunderstandings about self-management and increasing doctor counseling to highlight the progressive nature of MASLD and the necessity of routine follow-up. Addressing these obstacles, the chance of developing cirrhosis, hepatocellular carcinoma, and cardiovascular problems may be decreased if adherence to treatment and prevention measures is improved [33,34,35,36,37].

Among the current study’s participants who attended hospital follow-up (n = 24, 61.5%), most of them (41.7%) reported using weight-reduction approaches as part of their prevention and management practices. This finding is consistent with previous studies, which often propose weight loss as the most effective non-pharmacological intervention for NAFLD. It has been shown that even minor weight loss of 5–10% can considerably reduce inflammation, liver fat accumulation, and, in certain cases, fibrosis [16,38,39,40,41].

## 5. Recommendations

To reduce MASLD in Saudi Arabia, strengthen patient education, promote lifestyle changes, address follow-up barriers, and integrate MASLD screening and education into primary care. Patient education programs should clarify misconceptions about MASLD’s risk factors and sequelae, including obesity, diabetes, hypertension, and cardiovascular disease, and its progressive nature. Physicians should encourage patients to get regular checkups and adopt evidence-based dietary and exercise changes. As weight loss of 5–10% remains key to management, collaboration among dietitians, physiotherapists, and other healthcare professionals is essential. Address patient self-management dependency, lack of counseling, and awareness of disease severity. Use reminder systems and follow-up calls to improve adherence. Younger individuals and those with diabetes, obesity, or hyperlipidemia may have more difficulty with a healthy lifestyle and need targeted interventions. Additional research on regional barriers and customized intervention programs is also necessary.

## 6. Limitations

The convenience sampling method can lead to selection bias, which may compromise the generalizability of the results. Like most cross-sectional studies, it does not establish a causal relationship between the variables examined. Self-reported data reliance may introduce recall bias. Future research should employ longitudinal designs to enhance robustness and offer a more thorough understanding of the factors affecting MASLD knowledge and attitudes.

## 7. Conclusions

This nationwide study found that Saudis know little about NAFLD and have many misconceptions about its risk factors and treatment. This lack of understanding, along with over-reliance on lifestyle changes and little official medical guidance, highlights the urgent need for a coordinated public health response. A comprehensive plan is needed to reduce the rising prevalence. MASLD screening and evidence-based instruction in primary care are the first steps. Physicians can discuss risk factors and disease progression earlier with these tools. Public health efforts should dispel myths and show that MASLD is a serious, progressive disease linked to obesity, diabetes, and heart disease.

## Figures and Tables

**Figure 1 healthcare-13-03322-f001:**
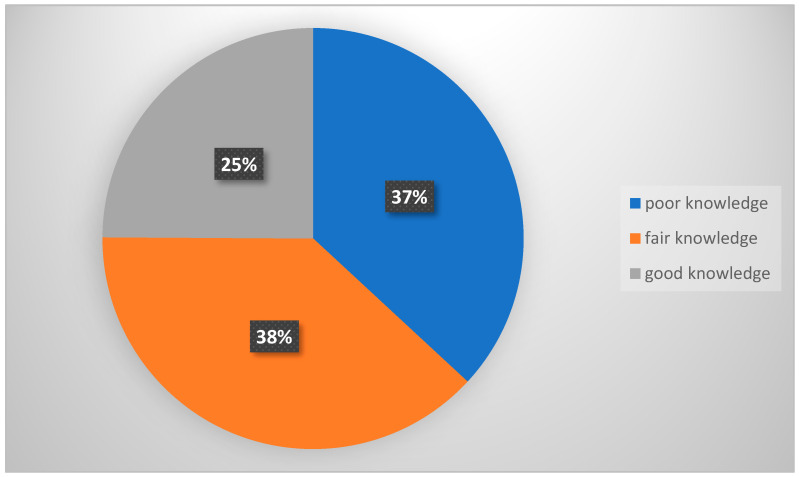
Participants’ knowledge levels about NAFLD.

**Table 1 healthcare-13-03322-t001:** Demographics of participants.

Variable	n (%)
Gender	
Male	231 (46)
Female	271 (54)
Age	
18–24	257 (51.2)
25–33	53 (10.6)
34–51	122 (24.3)
52–64	57 (11.4)
>65	9 (1.8)
Job	
Employer	172 (34.3)
Unemployed	45 (9.0)
Students	240 (47.8)
Retired	45 (9.0)
Education level	
High school	88 (17.5)
University	384 (76.3)
Postgraduate	30 (6.0)
body index	
below normal	36 (7.2)
Normal	230 (45.8)
Overweight	148 (29.5)
Obesity	34 (6.8)
I don’t know	54 (10.8)

**Table 2 healthcare-13-03322-t002:** Participants’ knowledge regarding NAFLD.

Variables	n (%)
Do you think you can get fatty liver without drinking alcohol?	
Yes	226 (45)
No	69 (13.7)
I don’t know	207 (41.2)
Have you ever heard of the term “non-alcoholic fatty liver disease” or “fatty liver”	
Yes	237 (47.2)
No	265 (52.8)
Do you think non-alcoholic fatty liver disease is a disease that requires hospital treatment?	
Agree	300 (59.8)
Disagree	27 (5.4)
I don’t know	175 (34.9)
NAFLD is life-threatening	
Yes	255 (50.8)
No	34 (6.8)
I don’t know	212 (42.2)
Methods for diagnosis	
Body mass index	40 (8)
Ultra sound	40 (8)
Blood test	49 (9.8)
All the above	236 (47.6)
I don’t know	133 (26.5)

**Table 3 healthcare-13-03322-t003:** Knowledge of participants about management of NAFLD.

Variables	n (%)
Change the diet (reduce sugars, processed foods, and saturated fats)	333 (66.3)
Physical activity	326 (64.9)
Controlling blood sugar levels for patients with diabetes, using medications if necessary	171 (34.3)
Vitamins	155 (30.9)
Lowering cholesterol and triglyceride levels through diet and medication if necessary	275 (54.8)
Eat a balanced diet rich in vegetables, fruits, and healthy proteins, with an emphasis on fiber and whole grains.	305 (60.8)

**Table 4 healthcare-13-03322-t004:** Association between participants’ knowledge levels and their demographics.

Variables	Knowledge Levels				95% Confidence Interval (CI)	*p*-Value
	Good n (%)	Fair n (%)	Poor n (%)	Total		
Gender						
Male	61 (26.4)	91 (39.4)	79 (34.2)	231	0.2 to 0.4	0.514
Female	64 (23.6)	101 (37.3)	106 (39.1)	271	0.18 to 0.4	
Age						
18–24	42 (16.3)	108 (42.0)	107 (41.6)	257	0.1 to 0.5	0.002 *
25–33	17 (29.8)	20 (35.1)	20 (35.1)	57	0.2 to 0.5	
34–51	43 (35.2)	38 (31.1)	41 (33.6)	122	0.2 to 0.4	
52–64	21 (36.8)	21 (36.8)	15 (26.3)	57	0.1 to 0.6	
65 and above	2 (22.2)	5 (55.6)	2 (22.2)	9	0.03 to 1.3	
Job						
Employer	58 (33.7)	55 (32.0)	59 (34.3)	172	0.24 to 0.44	0.001 *
Unemployed	14 (31.1)	18 (40.0)	13 (28.9)	45	0.15 to 0.63	
Students	38 (15.8)	99 (41.3)	103 (42.9)	240	0.11 to 0.52	
Retired	15 (33.3)	20 (44.4)	10 (22.2)	45	0.11 to 0.69	
Education level						
High school	23 (26.1)	37 (42.0)	29 (31.8)	88	0.16 to 0.58	0.008 *
University	87 (22.7)	145 (37.8)	152 (39.6)	384	0.18 to 0.46	
Postgraduate	15 (50.0)	10 (33.3)	5 (16.7)	30	0.05 to 0.82	
Body mass index						
Below normal	5 (13.9)	15 (41.7)	16 (44.4)	36	0.05 to 0.72	0.230
Normal	52 (22.6)	88 (38.3)	90 (39.1)	230	0.16 to 0.48	
Overweight	45 (30.4)	56 (37.8)	47 (31.8)	148	0.22 to 0.49	
Obesity	13 (38.2)	11 (32.4)	10 (29.4)	34	0.14 to 0.58	
I don’t know	10 (18.5)	22 (40.7)	22 (40.7)	54	0.09 to 0.62	

* *p* < 0.05 (Chi-square test).

**Table 5 healthcare-13-03322-t005:** Participant’s attitude towards NAFLD.

Variables	Yes n (%)	No n (%)
Do you believe that obesity causes NAFLD	243 (48.4)	259 (51.6)
Do you believe that NAFLD is caused by diabetes?	146 (29.1)	356 (70.9)
Do you believe that hypertension affects NAFLD?	120 (23.9)	382 (76.1)
Do you believe that liver cancer can be caused by NAFLD?	215 (42.8)	287 (57.2)
Do you believe that high blood cholesterol cause NAFLD?	260 (51.8)	242 (48.2)
Do you think that NAFLD can cause cardiovascular diseases	149 (29.7)	353 (70.3)

**Table 6 healthcare-13-03322-t006:** Association between Participants’ attitude levels and their demographics.

Variables	Knowledge Levels				95% Confidence Interval (CI)	*p*-Value
	Good n (%)	Fair n (%)	Poor n (%)	Total		
Gender						
Male	64 (27.7)	83 (35.9)	84 (36.4)	231	0.23 to 0.45	0.520
Female	68 (25.1)	91 (33.6)	112 (41.3)	271	0.19 to 0.49	
Age						
18–24	72 (28.0)	82 (31.9)	103 (40.1)	257	0.22 to 0.49	0.614
25–33	12 (21.1)	23 (40.4)	22 (38.6)	57	0.11 to 0.61	
34–51	26 (21.3)	47 (38.5)	49 (40.2)	122	0.14 to 0.51	
52–64	19 (33.3)	18 (31.6)	20 (35.1)	57	0.18 to 0.54	
65 and above	3 (33.3)	4 (44.4)	2 (22.2)	9	0.07 to 1.1	
Job						
Employer	43 (25.0)	62 (36.0)	67 (39.0)	172	0.18 to 0.49	0.725
Unemployed	9 (20.0)	17 (37.8)	19 (42.2)	45	0.09 to 0.66	
Students	70 (29.2)	76 (31.7)	94 (39.2)	240	0.23 to 0.47	
Retired	10 (22.2)	19 (42.2)	16 (35.6)	45	0.11 to 0.58	
Education level						
High school	26 (29.5)	33 (37.5)	29 (33.0)	88	0.19 to 52	0.261
University	94 (24.5)	132 (34.4)	158 (41.1)	384	0.19 to 0.48	
Postgraduate	12 (40.0)	9 (30.0)	9 (30.0)	30	0.13 to 0.69	
Body mass index						
Under normal	8 (22.2)	8 (22.2)	20 (55.6)	36	0.01 to 0.85	0.084
Normal	62 (27.0)	73 (31.7)	95 (41.3)	230	0.21 to 0.51	
Overweight	43 (29.1)	57 (38.5)	48 (32.4)	148	0.21 to 0.49	
Obesity	11 (32.4)	14 (41.2)	9 (26.5)	34	0.12 to 0.69	
I don’t know	8 (14.8)	22 (40.7)	24 (44.4)	54	0.06 to 0.66	

**Table 7 healthcare-13-03322-t007:** Participants’ responses to the management status and obstacles to NAFLD management.

Variables	n (%)
Diagnosis with NAFLD	
Yes	39 (7.8)
No	453 (92.2)
During your hospital visit, have you been recommended to change your lifestyle?	
Yes	29 (74.4)
No	10 (25.6)
Did you visit the hospital for more tests and management of NAFLD?	
Yes (go to the prevention/management question)	24 (61.5)
No (go to the reason question)	15 (38.5)
Your reason for not following-up with another hospital visit? (Multiple answers allowed) (n = 15)	
Not considered fatty liver a serious disease.	5 (33.3)
I believed that by changing my lifestyle on my own, I could control the illness (weight management, exercise management, etc.).	7 (46.7)
Lack of time to visit the hospital.	2 (13.3)
The cost of medical care.	4 (26.7)
My doctor has never advised me that I require illness management.	5 (46.7)
Prevention/management of NAFLD (Multiple answers allowed) (n = 24)	
I am not managing my NAFLD in any specific way.	8 (33.3)
Supplements for hyperlipidemia and the liver are available at pharmacies or online.	5 (20.8)
Hospital-prescribed drugs for hyperlipidemia and liver.	6 (25)
Reduction in calorie intake.	5 (20.8)
Increase in the amount of exercise.	7 (29.2)
Weight loss.	10 (41.7)

**Table 8 healthcare-13-03322-t008:** Perception of participants about the most important for the effective long-term management of non-alcoholic fatty liver disease.

Variables	n (%)
What do you think is the most important aspect of managing long-term non-alcoholic fatty liver disease? (Multiple answers)	
Make time for lifestyle changes	320 (63.7)
Health with treatment costs	221 (44)
Providing nutritional advice and periodic management by a nutritionist	286 (57)
Providing advice on how to exercise and periodic management by a sports specialist	232 (46.2)
Instructions on proper diet and exercise provided by a physician	283 (56.4)
If there is a mobile app for the prevention or management of non-alcoholic fatty liver disease, would you be willing to participate?	
Strongly agree	172 (34.3)
Agree	47 (9.4)
Neutral	122 (24.3)
Disagree	61 (12.2)
Strongly disagree	100 (19.9)
If there is a program to visit public health centers to prevent or manage non-alcoholic fatty liver disease, would you be willing to participate?	
Willing to actively participate	172 (34.3)
Willing to participate	47 (9.4)
Neutral	122 (24.3)
Little interest in participating	61 (12.2)
No interest in participating	100 (19.9)

## Data Availability

The original contributions presented in this study are included in the article and its Appendix A. Further inquiries can be directed to the corresponding author.

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
