# Peer review of "Bridging the Knowledge Gap: A National Survey on MASLD Awareness and Management Barriers in the Saudi Population"

_healthcare, 2025, doi:10.3390/healthcare13243322_

Round 1
Reviewer 1 Report
Comments and Suggestions for Authors
I reviewed the manuscript entitled "Bridging the Knowledge Gap: A National Survey on NAFLD Awareness and Management Barriers in the Saudi Population". You performed a systematic online self-administered questionnaire aiming to assess the level of public knowledge and attitudes related to NAFLD, in order to lay the ground for future educational and preventive strategies. This is a relevant topic, especially given rising NAFLD prevalence in the region. However, there are several aspects that need to be addressed before the publication of this manuscript. My concerns are listed below:
Major concerns
- There are numerous inconsistencies regarding the text, tables, and respectivelly figures regarding the numerical results. Please address these by rewriting the abstract, results and discussion sections:
- NAFLD - in the abstract section “58% had heard of NAFLD” (page 1). In the manuscript section, Table 2 (page 5): “Have you ever heard of the term “non-alcoholic fatty liver disease” or “fatty liver” Yes 210 (41.8%), No 292 (58.2%)”. In the manuscript section, Discussion (page 11): “just 47.2% of participants said they had heard of NAFLD”
- In the manuscript section, Table 1 (page 4):Students: 240 (47.8%), Employer: 172 (34.3%), Retired: 45 (9%), Unemployed: 45 (9%). This is not a correct interpretation of the table into the text. Please correct accordingly.
- There are discrepancies between the BMI in the text and in the table. In Page 3–4 you state that: “The proportion of individuals classified inside the normal weight category was 7.2%”. However, in Table 1 you state that Normal BMI = 45.8%, Below normal = 7.2%. Please align this paragraph with the table.
- There are problems with the terminology used. For instance, on page 6 the pie chart shows: Poor: 37%, Fair: 38% , Good: 25%. Throughout the text you use terms such as “low understanding”, “poor knowledge”, “satisfactory knowledge”. Please use the same terminology to avoid confusion.
Minor concerns - There are several typographical errors throughout the text. Please ensure that these are corrected.
- Page 6. Table 4. Table 2 Association between knowledge levels and demographics of the participants. Please correct
- Page 8. Table 5. Table 2: Association between attitude levels and demographics of the participants. Please correct
- Several tables lack footnotes. Please correct.
Author Response
Response to reviewers’ comments and amendments
Reviewer 1:
I reviewed the manuscript entitled "Bridging the Knowledge Gap: A National Survey on NAFLD Awareness and Management Barriers in the Saudi Population". You performed a systematic online self-administered questionnaire aiming to assess the level of public knowledge and attitudes related to NAFLD, in order to lay the ground for future educational and preventive strategies. This is a relevant topic, especially given rising NAFLD prevalence in the region. However, there are several aspects that need to be addressed before the publication of this manuscript. My concerns are listed below:
Major concerns
There are numerous inconsistencies regarding the text, tables, and respectively figures regarding the numerical results. Please address these by rewriting the abstract, results and discussion sections:
Comment: NAFLD - in the abstract section “58% had heard of NAFLD” (page 1). In the manuscript section, Table 2 (page 5): “Have you ever heard of the term “non-alcoholic fatty liver disease” or “fatty liver” Yes 210 (41.8%), No 292 (58.2%)”. In the manuscript section, Discussion (page 11): “just 47.2% of participants said they had heard of NAFLD”
Response: Thank you for your valuable comment and amendment. You might find that all numbers and percentages are consistent throughout the revised manuscript, whether in tables or in the text.
Comment: In the manuscript section, Table 1 (page 4): Students: 240 (47.8%), Employer: 172 (34.3%), Retired: 45 (9%), Unemployed: 45 (9%). This is not a correct interpretation of the table into the text. Please correct accordingly.
Response: Thank you for your valuable comment and amendment. You might find that all numbers and percentages are consistent throughout the revised manuscript, whether in tables or in the text.
Comment: There are discrepancies between the BMI in the text and in the table. In Page 3–4 you state that: “The proportion of individuals classified inside the normal weight category was 7.2%”. However, in Table 1 you state that Normal BMI = 45.8%, Below normal = 7.2%. Please align this paragraph with the table.
Response: Thank you for your valuable comment and amendment. Aligned as directed, you might find that all numbers and percentages are consistent throughout the revised manuscript, whether in tables or in the text.
Comment: There are problems with the terminology used. For instance, on page 6 the pie chart shows: Poor: 37%, Fair: 38% , Good: 25%. Throughout the text you use terms such as “low understanding”, “poor knowledge”, “satisfactory knowledge”. Please use the same terminology to avoid confusion.
Response: Thank you for your valuable comment and amendment. Unified as directed, and you might find that all numbers and percentages are consistent throughout the revised manuscript, whether in figures or in the text.
Minor concerns
There are several typographical errors throughout the text. Please ensure that these are corrected.
Comment: Page 6. Table 4. Table 2 Association between knowledge levels and demographics of the participants. Please correct
Response: Thank you for your valuable comment and amendment. Corrected as directed.
Comment: Page 8. Table 5. Table 2: Association between attitude levels and demographics of the participants. Please correct
Response: Thank you for your valuable comment and amendment. Corrected as directed.
Comment: Several tables lack footnotes. Please correct.
Response: Thank you for your valuable comment and amendment. All table footnotes are added whenever applicable.

Reviewer 2 Report
Comments and Suggestions for Authors
This is a good topic that deals with public knowledge and barriers to NAFLD management in Saudi Arabia. The nationwide approach and relatively large sample (n=502) add robustness and relevance to the findings. The study offers valuable insights that can guide future health education and public policy initiatives, so I think It is a very very good study. I have som e issues:
Please look at the language again. Some section particularly in the Methods and Discussion, contain awkward phrasing and repetitive wording. Get some profesionall english editign.
Please clarify whether the questionnaire was validated or pilot-tested before use
please prrovide Cronbach’s alpha or another measure of internal consistency for the knowledge and attitude sections
please dscribe sampling technique in more detail (convenience vs. random sampling)
include confidence intervals where applicable
figures could be improved for readability, Figure 1 (?)
That is all
Author Response
Reviewer 2
This is a good topic that deals with public knowledge and barriers to NAFLD management in Saudi Arabia. The nationwide approach and relatively large sample (n=502) add robustness and relevance to the findings. The study offers valuable insights that can guide future health education and public policy initiatives, so I think It is a very very good study. I have som e issues:
Comment: Please look at the language again. Some sections, particularly in the Methods and Discussion, contain awkward phrasing and repetitive wording. Get some professional English editing.
Response: Thank you for your valuable comment and amendment. The authors will send the approved revised manuscript to the journal’s expert editors for professional Language and Figure Editing.
Comment: Please clarify whether the questionnaire was validated or pilot-tested before use.
Response: Thank you for your valuable comment and amendment. In the revised manuscript, it was written “The study adopted a validated, systematic, self-administered online questionnaire modified from previous studies”.
Comment: Please provide Cronbach’s alpha or another measure of internal consistency for the knowledge and attitude sections
Response: Thank you for your valuable comment and amendment. In the revised manuscript, we added the assessment method of the internal consistency of the knowledge and attitude tiers using Cronbach's alpha.
Comment: Please describe sampling technique in more detail (convenience vs. random sampling)
Response: Thank you for your valuable comment and amendment. In the revised manuscript, we added the sampling technique and calculations.
Comment: Include confidence intervals where applicable
Response: Thank you for your valuable comment and amendment. In the revised manuscript, we added the CI whenever applicable (Tables 4 and 6).
Comment: The figures could be improved for readability, Figure 1
Response: Thank you for your valuable comment and amendment. The authors will send the approved revised manuscript to the journal’s expert editors for professional Language and Figure Editing.

Reviewer 3 Report
Comments and Suggestions for Authors
In the work “Bridging the Knowledge Gap: A National Survey on NAFLD Awareness and Management Barriers in the Saudi Population”, authors have presented the importance of awareness about NAFLD in the Kingdom of Saudi population. The authors have used appropriate statistical analysis to validate their data involved in the determination of the knowledge gap and awareness in the management and prevention of the NAFLD. There are few comments raised while reviewing the manuscript. It’s minor revision. I suggest that the authors address the following comments in the revised manuscript.
- I would suggest the authors to use metabolic-dysfunction associated lifer disease (MAFLD), instead of NAFLD throughout the manuscript. The term MAFLD is more appropriate and reflection of underlying pathogenesis for this liver disease.
- On page no. 2 of 15, the authors have mentioned the statement, “currently, there are no FDA-approved drugs for NAFLD. However, recently, the USF FDA granted accelerated approval to resmetirom (Rezdiffra™), liver-directed thyroid hormone receptor-beta (THR-β) agonist. This is the first and approved drug for the treatment of metabolic dysfunction-associated steatohepatitis (MASH) with moderate to advanced liver fibrosis. I suggest the authors to include this in the introduction part with an appropriate references.
- I suggest the authors to include a flow chart in the revised manuscript to clearly outline the inclusion and exclusion criteria used for the selection of the participants in the present study.
- The in-text citations for Table 1 and 2 are missing in the present manuscript.
- Please provide complete and informative caption for Table 2. Maintain a uniform format across all tables. The authors used n (%) in Table 3. I suggest the authors follow the same for all Tables.
- Please check the typographical errors in the Tables head 4 and 5.
- There are numerous inconsistency and errors in the Tables numbering and their continuity in the manuscript. Please thoroughly check and correct Table 3 on page 7 of 15, Table 5 on page 8 of 15, Table 5 on page 9 of 15, and Table 6 on page 10 of 15.
- Please clearly state the limitation of the study in the revised manuscript.
- General comments: There are many typographical errors. I suggest the authors to verify once again all the typographical errors in the Tables continuity, numbering, and in-text citations throughout the manuscript. Finally, check the general English grammars throughout the manuscript.
Author Response
Reviewer 3
In the work “Bridging the Knowledge Gap: A National Survey on NAFLD Awareness and Management Barriers in the Saudi Population”, authors have presented the importance of awareness about NAFLD in the Kingdom of Saudi population. The authors have used appropriate statistical analysis to validate their data involved in the determination of the knowledge gap and awareness in the management and prevention of the NAFLD. There are few comments raised while reviewing the manuscript. It’s minor revision. I suggest that the authors address the following comments in the revised manuscript.
Comment: I would suggest the authors to use metabolic-dysfunction associated lifer disease (MAFLD), instead of NAFLD throughout the manuscript. The term MAFLD is more appropriate and reflection of underlying pathogenesis for this liver disease.
Response: Thank you for your valuable comment and amendment. Although your suggestions are both valuable and up-to-date, the study's population consists of the public living in Saudi Arabia, who find it easier to become familiar with NAFLD than with MAFLED. Actually, your suggestion encouraged the authors to propose a new cross-sectional survey where primary care physicians or gastroenterologists would be targeted.
Comment: On page no. 2 of 15, the authors have mentioned the statement, “currently, there are no FDA-approved drugs for NAFLD. However, recently, the USF FDA granted accelerated approval to resmetirom (Rezdiffra™), a liver-directed thyroid hormone receptor-beta (THR-β) agonist. This is the first and approved drug for the treatment of metabolic dysfunction-associated steatohepatitis (MASH) with moderate to advanced liver fibrosis. I suggest the authors to include this in the introduction part with an appropriate references.
Response: Thank you for your valuable comment and amendment. The US FDA’s announcement of the new medication was added and cited accordingly.
Comment: I suggest that the authors include a flow chart in the revised manuscript to clearly outline the inclusion and exclusion criteria used for the selection of the participants in the present study.
Response: Thank you for your valuable comment and amendment. Since the study's population consists of the public living in Saudi Arabia, and the questionnaire was systematic and self-administered online, it would be better to stick to the methodology mentioned because in a cross-sectional online survey, there is no conventional "enrollment" from a pool of evaluated persons as seen in a cohort study. The flowchart usually starts with "Total clicks on a survey link" or "Total page views." These numbers can be inaccurate and inflated by bots or people who are just curious. It doesn't reveal how many individuals in your target group never saw the link (for example, all social media users who didn't see your ad or all email list members who deleted the message).
Comment: Please provide complete and informative caption for Table 2. Maintain a uniform format across all tables. The authors used n (%) in Table 3. I suggest the authors follow the same for all Tables. Please check the typographical errors in the Tables head 4 and 5.
Response: Thank you for your valuable comment and amendment. Completed and unified as directed.
Comment: There are numerous inconsistency and errors in the Tables numbering and their continuity in the manuscript. Please thoroughly check and correct Table 3 on page 7 of 15, Table 5 on page 8 of 15, Table 5 on page 9 of 15, and Table 6 on page 10 of 15.
Response: Thank you for your valuable comment and amendment. Consistency is strictly followed, and you might observe that all numbers and percentages are consistent throughout the revised manuscript, whether in tables or in the text.
Comment: Please clearly state the limitation of the study in the revised manuscript.
Response: Thank you for your valuable comment and amendment. Done as directed.
Comment: General comments: There are many typographical errors. I suggest that the authors verify once again all the typographical errors in the Tables continuity, numbering, and in-text citations throughout the manuscript. Finally, check the general English grammar throughout the manuscript.
Response: Thank you for your valuable comment and amendment. The authors will send the approved revised manuscript to the journal’s expert editors for professional Language and Figure Editing.

Round 2
Reviewer 1 Report
Comments and Suggestions for Authors
The authors followed the instructions.
Author Response
Greetings, Academic Editor at Healthcare Journal Thank you for your efforts. I, on behalf of the authors, appreciate your ongoing efforts to update the journal's content, including articles and other manuscripts. From this point, I want to inform you that all amendments to update NAFLD to MASLD were made throughout the article, with an explanation of the rationale for keeping the questionnaire and results using the term NAFLD in the study tool section of the methodology. We cited a recent reference in the introduction, "From NAFLD to MASLD: Updated naming and diagnosis criteria for fatty liver disease," and renumbered the references accordingly. REGARDS
